



# Evaluation of version 3.0B of the BEHR OMI NO$_2$ product

Joshua L. Laughner[1], Qindan Zhu[2], and Ronald C. Cohen[1,2]

[1]Department of Chemistry, University of California, Berkeley, Berkeley, CA 94720
[2]Department of Earth and Planetary Sciences, University of California, Berkeley, Berkeley, CA 94720

**Correspondence:** Ronald C. Cohen (rccohen@berkeley.edu)

**Abstract.** Version 3.0B of the Berkeley High Resolution (BEHR) OMI NO$_2$ product is designed to accurately retrieve daily variation in the high spatial resolution mapping of tropospheric column NO$_2$. To assess the product, we compare against in situ aircraft profiles and Pandora vertical column densities (VCDs). We also compare the WRF-Chem simulation used to generate the a priori NO$_2$ profiles against observations. We find that using daily NO$_2$ profiles improves the VCDs retrieved

in urban areas relative to low resolution or monthly a priori by amounts that are large compared to current uncertainties in NO$_x$ emissions and chemistry (of order 10% to 30%). Based on this analysis, we offer suggestions to consider when designing retrieval algorithms and validation procedures for upcoming geostationary satellites.

## 1 Introduction

NO$_x$ ($\equiv$ NO + NO$_2$) is an atmospheric trace gas emitted by anthropogenic activity (predominantly combustion, e.g. motor

vehicles and power plants), lightning, biomass burning, and soil microbes. It plays an important role in air quality, as a major controlling factor in ozone and aerosol production, as well as being toxic itself.

Satellite observations of NO$_2$ have proven to be extremely useful in constraining anthropogenic (e.g. Richter et al., 2005; Kim et al., 2006, 2009; van der A et al., 2008; Konovalov et al., 2010; Russell et al., 2010; Beirle et al., 2011; Castellanos and Boersma, 2012; Russell et al., 2012; Zhou et al., 2012; McLinden et al., 2014; Lu et al., 2015; Liu et al., 2016, 2017; Miyazaki

et al., 2012, 2017), lightning (e.g. Beirle et al., 2004; Martin et al., 2007; Beirle et al., 2010; Bucsela et al., 2010; Miyazaki et al., 2014; Pickering et al., 2016; Nault et al., 2017), soil (e.g. Bertram et al., 2005; van der A et al., 2008; Hudman et al., 2010, 2012; Zörner et al., 2016), and biomass burning (e.g. Mebust et al., 2011; Huijnen et al., 2012; Mebust and Cohen, 2013, 2014; Bousserez, 2014; Schreier et al., 2014; Castellanos et al., 2015; van Marle et al., 2017) emissions.

Satellite observations of NO$_2$ relate absorption of light in the $\sim$ 400–460 nm range of reflected Earthshine radiances to a

total column measurement of NO$_2$ using differential optical absorption spectroscopy (DOAS, Boersma et al., 2001; Richter and Wagner, 2011) or a similar technique (e.g. van Geffen et al., 2015). Most applications of satellite NO$_2$ observations to constrain emissions or otherwise study air quality are focused on the tropospheric contribution to the total column; therefore the stratospheric column must be removed. Several methods have been implemented to do so (e.g. Boersma et al., 2007; Bucsela et al., 2013). The tropospheric slant column density (SCD) is then converted to a vertical column density (VCD) through the

use of an air mass factor (AMF, McKenzie et al., 1991; Slusser et al., 1996; Burrows et al., 1999; Palmer et al., 2001) that accounts for the effect of path length, surface reflectivity and elevation, NO$_2$ vertical distribution, clouds, and aerosols.





There have been numerous studies evaluating OMI $NO_2$ products against in situ aircraft profiles and ground based column measurements. This is not meant to be an exhaustive list, but to provide a summary of the results of evaluations of existing standard OMI $NO_2$ products.

The first-generation NASA Standard Product (SP) and KNMI DOMINO products were evaluated by Bucsela et al. (2008)

and Hains et al. (2010) using aircraft profiles from multiple campaigns and Russell et al. (2011) using an extrapolation method with ARCTAS-CA aircraft data. These studies all identified a high bias in the DOMINO VCDs; by comparing the DOMINO a priori profiles to aircraft and lidar profiles Hains et al. (2010) found evidence that this was caused by insufficient vertical mixing in the DOMINO a priori profiles, which was corrected in DOMINO v2.

Lamsal et al. (2014) undertook a detailed evaluation of the NASA SP v2, primarily focusing on data from the Deriving Infor-

mation on Surface Conditions from COlumn and VERtically Resolved Observations Relevant to Air Quality (DISCOVER-AQ) campaign in Baltimore, MD, USA. This work combined evaluation of the a priori profile against aircraft measurements along with validation of OMI VCDs with aircraft and ground-based VCDs. They found that the NASA SP v2 VCDs were generally biased low in urban areas and high in rural or suburban areas. This is consistent with the effect of coarse a priori profiles (Russell et al., 2011); in a large urban area like the Baltimore/Washington D.C. urban corridor, a coarse profile can capture the

average urban characteristic profile, but on the edge, a coarse profile cannot capture the transition from urban to rural.

Krotkov et al. (2017) and Goldberg et al. (2017) both evaluated the NASA SP v3, primarily using ground based VCD observations. They found it to be biased low by $\sim 50\%$ in the Baltimore area (Goldberg et al., 2017) and $50\%$ or more Hong Kong (Krotkov et al., 2017), but better than SP v2 in remote areas, due to the improved total column fitting implemented in version 3. Ialongo et al. (2016) also compared versions 2 and 3 of the NASA SP and version 2 of DOMINO against ground

based column measurements in Helsinki, one of only a few studies at high latitudes ($> 60°$). They found that SP v3 was biased 30% low, while the version 2 products were not. They attributed this to cancellation of errors in the version 2 products, namely the high bias in the total OMI columns corrected by van Geffen et al. (2015), and the representativeness mismatch between OMI pixels and Pandora measurements.

Here we present an evaluation of version 3.0B of the BErkeley High Resolution (BEHR) OMI $NO_2$ retrieval. Version 3.0B

implements several changes over v2.1C:

- Daily profiles for selected years

- Updated 12 km WRF-Chem $NO_2$ profiles with a more complete chemical mechanism (Zare et al., 2018), updated anthropogenic emissions, lightning $NO_x$ emissions added

- Use of v3.0 NASA Standard Product (SP) tropospheric SCDs

- Directional surface reflectance

- Variable tropopause height

- Surface pressure combining a high resolution terrain database with WRF-simulated surface pressure





The motivation for this upgrade stems from ideas developed in Laughner et al. (2016), where we showed that daily, high resolution a priori profiles are necessary for a retrieval to simultaneously retrieve $NO_x$ VCDs and lifetime to accuracies better than 30%. As our goal is to study the relationship between changes in $NO_x$ VCDs/emissions and $NO_x$ lifetime across the US, and resolving open questions requires higher relative precision and high accuracy than prior retrievals, we have developed a

new product with daily 12 km a priori profiles. Therefore, in this work, we first evaluate the simulated WRF-Chem profiles against aircraft measurements and OMI SCDs to demonstrate that the daily profiles accurately represent the real atmosphere. We then directly evaluate the retrieved VCDs using both aircraft and Pandora observations and show that v3.0 is generally superior to v2.1C, and that using daily profiles improves the overall quality of the retrieval.

## 2  Methods: models and observations

### 2.1  BEHR

The BEHR OMI $NO_2$ retrieval is described in detail in Laughner et al. (2018f). Briefly, the BEHR retrieval calculates a tropospheric air mass factor (AMF) using high resolution a priori input data for surface reflectance, surface elevation, and $NO_2$ vertical profiles; the $NO_2$ profiles are simulated with WRF-Chem (Sect. 2.2). To capture the day-to-day variation in $NO_2$ profiles, daily profiles are used. Currently, 2005, 2007–2009, and 2012–2014 are available. Other years will be posted as

processing is completed. A second subproduct uses monthly average profiles (simulated for 2012) to retrieval all years of the OMI data record.

The BEHR AMF is used to convert the tropospheric slant column densities (SCDs) available in the NASA OMI $NO_2$ standard product to tropospheric vertical column densities (VCDs). For full details of the AMF calculation, see Laughner et al. (2018f). The BEHR product is available for download as HDF version 5 files at behr.cchem.berkeley.edu.

### 2.2  WRF-Chem

The WRF-Chem model version used to simulate the a priori $NO_2$ profiles for BEHR v3.0B is v3.5.1 (Grell et al., 2005). The model domain is 405 (east-west) by 254 (north-south) 12 km grid cells centered on 39° N 97° W with 29 vertical levels. Meterological initial, boundary, and nudging conditions are taken from the North American Regional Reanalysis (NARR) product; boundary conditions and four-dimension data analysis (FDDA) nudging (Liu et al., 2006) is applied every 3 hours.

Temperature, water vapor, and U/V winds are nudged with nudging coefficients of 0.0003 s$^{-1}$.

The chemical mechanism used is described in Zare et al. (2018), which has a very detailed description of alkyl nitrate and nighttime chemistry. Methyl peroxynitrate (MPN) chemistry was added (Browne et al., 2011) to improve upper tropospheric chemistry. Anthropogenic emissions are from the National Emissions Inventory, 2011, scaled by EPA annual total emissions (EPA, 2016) to the model year. Biogenic emissions are from the Model for Emissions of Gases and Aerosols from Nature

(Guenther et al., 2006). Lightning emissions are parameterized following Laughner and Cohen (2017) for a simulation with FDDA active (500 mol NO flash$^{-1}$, 2x base flashrate).



| Data field | Condition |
|---|---|
| XTrackQualityFlags | Must be 0 |
| VcdQualityFlags | Must be an even number |
| CloudFraction | Must be $\leq 0.2$ |
| BEHRAMFTrop | Must be a non-fill value $> 10^{-6}$ |

**Table 1.** Criteria that OMI pixels must meet to be used in any comparison.

Chemical initial and boundary conditions are interpolated to the WRF grid using the MOZBC utility (https://www2.acom.ucar.edu/wrf-chem/wrf-chem-tools-community). For 2007 and later model years, chemical data is obtained from the MOZART model runs available at https://www.acom.ucar.edu/wrf-chem/mozart.shtml. For 2005 and 2006, chemical data is obtained from a GEOS-Chem model run, described in Laughner et al. (2018f).

## 2.3 Pandora ground-based columns

Evaluation of satellite $NO_2$ VCDs usually uses one of two methods. First, total satellite columns can be directly compared to a ground-based column measurement, such as a Pandora spectrometer (Herman et al., 2009) or multi-axis DOAS (MAX-DOAS) instrument (Hönninger et al., 2004). In the case of a direct-sun measurement, such as a Pandora spectrometer, the AMF required is only a geometric AMF to account for the path length difference between the slant and vertical column, since the multiple scattering that necessitates the use of a more complex AMF in the satellite retrieval is a much smaller signal than the direct-sun signal (Herman et al., 2009).

We compare against Pandora ground based column measurements taken during the four DISCOVER-AQ campaigns. For each OMI overpass, pixels are matched with Pandora sites that lie within the pixel boundaries defined by the FoV75 corners in the OMPIXCOR product (Kurosu and Celarier, 2010). Only pixels meeting the criteria in Table 1 are used. If multiple valid pixels from the same overpass encompass the Pandora site, their VCDs are averaged. As in Goldberg et al. (2017), the stratospheric VCD from the NASA Standard Product is added to the tropospheric VCD to obtain a total column, since the Pandora columns do not separate stratospheric and tropospheric contributions.

Pandora observations are matched in time to the OMI observations using the exact time of observation for each pixel given in the OMI data files. As in Goldberg et al. (2017), Pandora observations $\pm 1$ h from the OMI observation are averaged.

## 2.4 In situ aircraft profiles

The other common method of evaluating satellite VCDs is to use in situ measurements of $NO_2$ by instrumented aircraft that flies a vertical profile to calculate a VCD by integrating the $NO_2$ concentrations vertically. Ideally, the aircraft should fly a spiral path that provides a complete vertical sampling of the troposphere over a ground footprint similar in scale to the satellite pixel; the DISCOVER-AQ campaigns held in Maryland, California, Texas, and Colorado between 2011 and 2014 were designed to provide this sampling over the lower troposphere. In other cases, the VCD calculated from integrating the aircraft profiles is





often matched to satellite pixels in which the boundary layer is sampled (e.g. Bucsela et al., 2008; Hains et al., 2010), on the assumption that upper troposphere (UT) sampling from adjacent pixels is sufficient.

We calculate tropospheric VCDs from in situ $NO_2$ profiles measured from aircraft. We use six campaigns: the four DISCOVER-AQ campaigns (https://www-air.larc.nasa.gov/missions/discover-aq/discover-aq.html) in Maryland (2011), California (2013),

Texas (2013), and Colorado (2014), the Southeast Nexus campaign (2013, southeast US, SENEX Science Team, 2013), and the Studies of Emissions and Atmospheric Composition, Clouds, and Climate Coupling by Regional Surveys (SEAC4RS, 2013, Toon et al., 2016). For the DISCOVER-AQ and SEAC4RS campaigns, we use 1 second $NO_2$ data from the TD-LIF instrument is used (Nault et al., 2015; Wooldridge et al., 2010; Day et al., 2002; Thornton et al., 2000). For the SENEX campaign, we use 1 second data from the chemiluminescence instrument (Ryerson et al., 1999).

We draw on methodology from several papers (Bucsela et al., 2008; Hains et al., 2010; Lamsal et al., 2014) for our approach. Similar to (Hains et al., 2010), only profiles with a minimum radar altitude $< 500$ m and at least 20 measurements below 3 km above ground level (AGL) are used. In the DISCOVER-AQ campaigns, individual profiles are demarcated in the data by a profile number. In the SENEX and SEAC4RS data, profiles were identified manually as periods when the aircraft was consistently ascending or descending. The profile measurements are binned to the same pressure levels used in the BEHR

algorithm and the final profile uses the median of each bin.

Profiles are spatially matched to OMI pixels if any of the 1 second measurements in the bottom 3 km AGL lies within the FoV75 pixel boundaries. As with Pandora data, OMI pixels must meet the criteria in Table 1 to be included; all VCDs from valid pixels intersecting the profile are averaged to yield a single VCD to compare against the profile. Only profiles with an mean observation time of all points in the bottom 3 km AGL within 1.5 h of the mean OMI observation time for the orbit are

used.

To calculate a VCD from the in situ measurements, the aircraft profiles are integrated from the average surface pressure to the average tropopause pressure of the matched pixels. The surface and tropopause pressure are used from the product being evaluated, i.e. aircraft profiles are integrated between BEHR surface and tropopause pressure for comparison with BEHR VCDs and NASA surface and tropopause pressures for comparison with NASA VCDs. For BEHR v2.1C comparisons, 200 hPa is

used as the fixed tropopause pressure. Aircraft profiles that do not span the necessary vertical extent are extended similarly to Lamsal et al. (2014). The aircraft profile is extended to the surface by using the ratio of modeled concentrations at each of the missing levels to the lowest level with aircraft data to scale the bottom bin with aircraft data. Missing profile levels above the top of the aircraft profile are replaced with model data. We use modeled $NO_2$ profiles from v9.02 of the GEOS-Chem global chemical transport model (Bey et al., 2001), with the updated chemistry and lightning emission rates described in Nault et al.

(2017). The $NO_2$ profiles are monthly averages of model output from 2012 sampled between 12:00 and 14:00 local standard time. We avoid using the a priori WRF-Chem profiles for this so that the aircraft VCDs are independent of the retrieved VCDs.

We also used the extrapolation method from Hains et al. (2010), where the median of the top 10 and bottom 10 points are extrapolated to the tropopause and surface pressures, respectively. The median of the top 10 points must be $< 100$ pptv. As in Hains et al. (2010), a detection limit of 3 pptv is assumed, and if the median to be extrapolated is less than 3 pptv, it is set to

one-half of the detection limit, 1.5 pptv.





In addition, we directly compare the a priori profiles to the in situ aircraft profiles. This is done as in Laughner and Cohen (2017); for each 1 second data point in the aircraft data, the nearest WRF-Chem output time is selected, and the model grid cell containing the aircraft location is sampled. This effectively samples the model output as if the aircraft were flying through the model world.

We use a similar set of aircraft campaigns here as for the VCD evaluation (Sect. 2.4); the only difference being that we use the Deep Convective Clouds and Chemistry (Barth et al., 2015) instead of SENEX. The DC3 campaign focused on outflow from convective systems (i.e. thunderstorms) and so is used to evaluate the lightning $NO_x$ parameterization. The DC3 campaign had better UT sampling but far fewer profiles than SENEX. The DISCOVER-AQ campaigns focused on satellite validation, flying repeated spirals over 6–8 sites during each campaign; however, for the average comparison, we use all data, not just that taken
during the spirals.

## 3   WRF-Chem profile evaluation

### 3.1   Comparison with in situ aircraft profiles

Figure 1 shows campaign averaged profiles matched with WRF profiles from the four DISCOVER-AQ campaigns, the DC3 campaign, and the SEAC4RS campaign. We compare the monthly average $NO_2$ profiles from BEHR v2.1C and v3.0B for all
campaigns, as well as the daily v3.0B profiles. The plots shown only use data between 12:00 and 15:00 local standard time, since the v3.0 monthly average profiles are calculated as a weighted average that only includes contributions from $\pm 1$ h from OMI overpass; this way all profiles get a fair comparison to the observations.

    In general, the v3.0 profiles show better agreement with observed profiles than the v2.1 profiles, except during the California DISCOVER-AQ campaign. The most dramatic example is the Maryland DISCOVER-AQ campaign, where the factor of $\sim 2$
reduction in $NO_2$ concentration (likely due to updating emissions from 2005 to 2012) brings the modeled profiles into substantially better agreement with the observed profiles. In the California DISCOVER-AQ campaign, the v2.1 profiles managed to capture an elevated layer of $NO_2$ that the v3.0 profiles did not; though we note that transport in California's central valley is notorious difficult to model (Hu et al., 2010, and references therein). In Texas, the v3.0 profiles and v2.1 profiles lie on opposite sides of the observed profiles, possibly suggesting that emissions in Houston did not decrease as much in fact as in the
NEI inventory driving the v3.0 WRF simulations. In Colorado, both the v3.0 and v2.1 profile match observations reasonably well. The daily profiles do a better job capturing the decrease in $NO_2$ between 750 and 600 hPa than the v3.0 monthly or v2.1 profiles; this may be due to day-to-day variability in recirculation from the upslope/downslope winds (e.g. Sullivan et al., 2016).

    We include the SEAC4RS and DC3 campaigns to check the simulation of lightning $NO_x$ in the profiles. The daily profiles
show similar agreement to the DC3 observations as in Laughner and Cohen (2017). Restricting the DC3 data to 12:00–15:00 local standard time as we have done here reduces the strength of the lightning signal, since the strongest lighting occurs after OMI overpass (Lay et al., 2007; Williams et al., 2000). Compared to Laughner and Cohen (2017), the discrepancy between modeled and observed profiles decreased around 500 hPa, increased around 400 hPa, and is similarly small around 200 hPa.





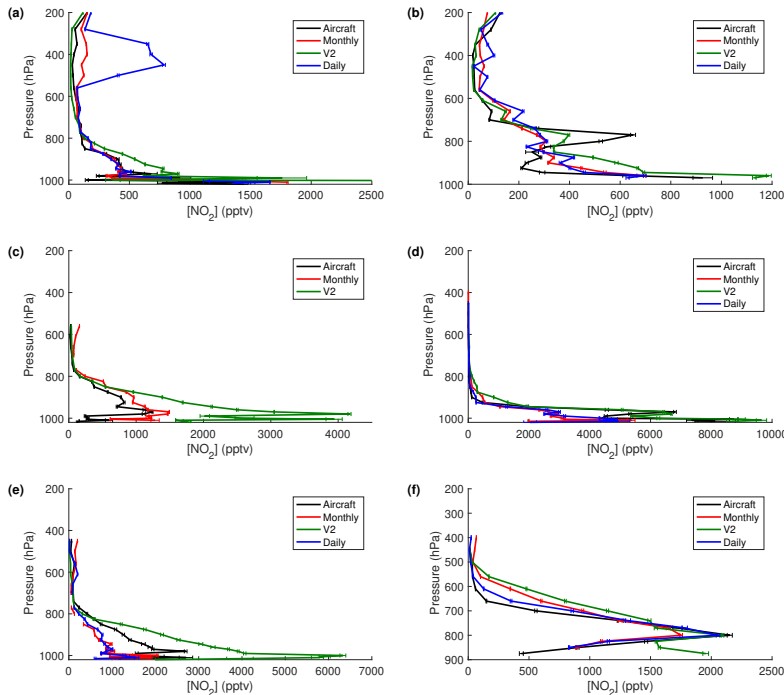

**Figure 1.** Comparison of average WRF-Chem and aircraft $NO_2$ profiles from the **(a)** SEAC4RS, **(b)** DC3, and DISCOVER-AQ campaigns, the latter in **(c)** Maryland, **(d)** California, **(e)** Texas, and **(f)** Colorado. Aircraft profiles are shown in black, BEHR v2.1 profiles in green, BEHR v3.0 monthly profiles in red, and (where available) BEHR v3.0 daily profiles in blue. The WRF and aircraft data are matched as described in Sect. 2.4 and binned by pressure. Uncertainties are 1 standard deviation of all profiles averaged. Note that for SEAC4RS the v2 profile reaches a maximum of $\sim 8000$ pptv, off the plot axes.

Surprisingly, the difference between the v2.1 and v3.0 profile around 200 hPa is not as significant as the difference between the lightning and no-lighting cases in Laughner and Cohen (2017). This is unexpected as the v2.1 profiles did not include lightning $NO_x$ emission. It is possible that convection of greater surface $NO_x$ concentrations is driving the v2.1 UT concentration.

The SEAC4RS campaign covers the southeast US, which has very active lightning (Hudman et al., 2007; Travis et al., 2016).
5   The daily profiles demonstrate a substantial overestimate in UT $NO_2$ (between 600 and 200 hPa). This is centered in the SE US; model-measurement discrepancies between 600 and 200 hPa in the rest of the country are $< 500$ pptv (not shown). As discussed in Laughner et al. (2018f), the southeast US exhibits greater $NO_2$ VCDs (and therefore smaller AMFs) when using daily profiles; that is opposite with the profiles seen here, as greater $NO_2$ at higher altitudes results in larger AMFs. Laughner et al. (2018f) showed that the 3 month average daily shape factor over the SE US had less contribution from UT $NO_2$ than
10  the monthly profiles; this indicates that on average pixels in the SE US are not influenced by lightning, but that the SEAC4RS



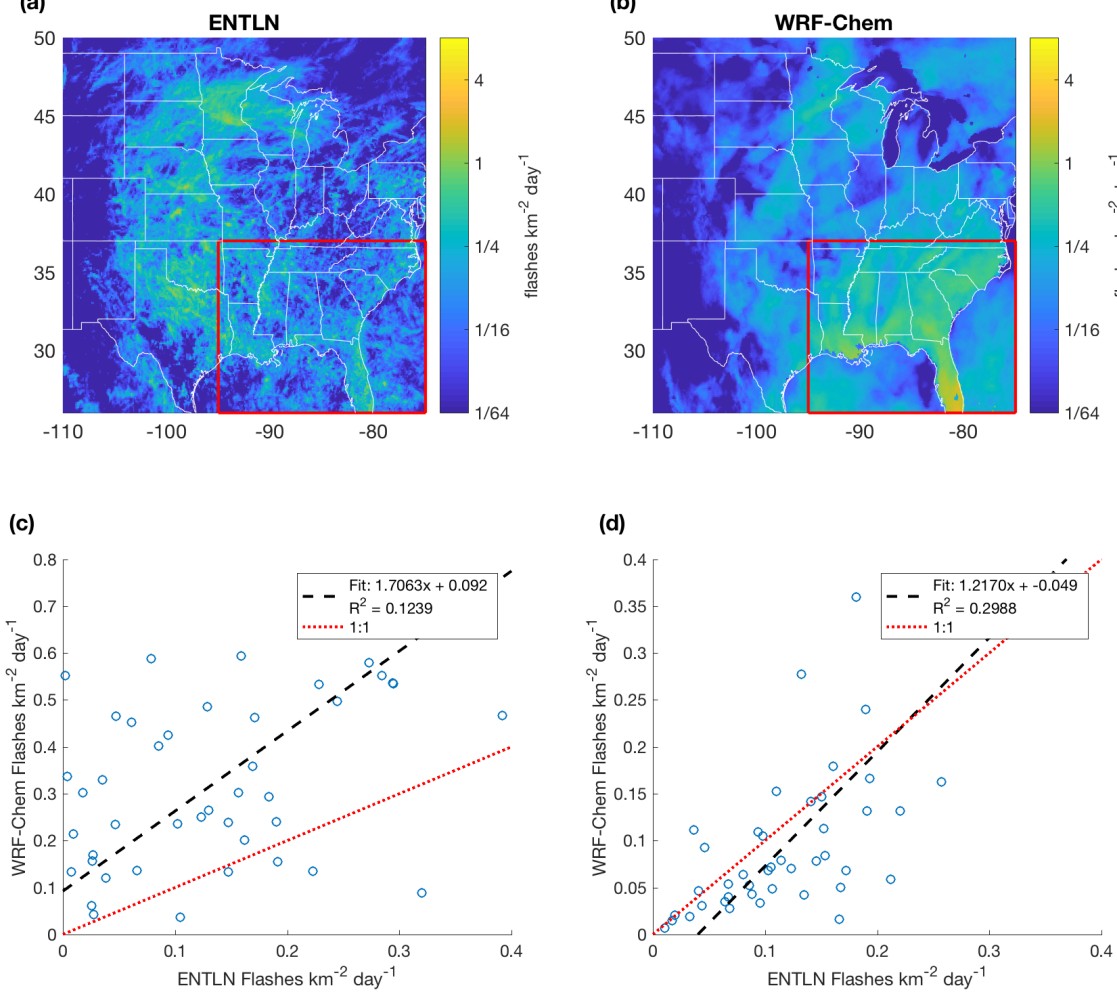

**Figure 2.** Comparison between observed and simulated flash density from May 13 to June 23 2012. **(a)** and **(b)** show the mean flash density averaged over the study period from ENTLN and WRF-Chem, respectively. Both are gridded at 12 km grid spacing. **(c)** and **(d)** show the correlation between total flash density per day between WRF and ENTLN in **(c)** the southeast US (denoted by the red box in **a** and **b**) and **(d)** elsewhere in CONUS.

sampling tended to select for convective outflow. However, this does indicate that the simulation of the UT in the southeast US is biased high.

To investigate the cause of this bias, we compare the WRF lightning flash density to that measured by the Earth Networks Total Lightning Network (ENTLN). ENTLN is a ground-based lightning observation network with more than 900 sensors deployed in the contiguous US. The sensors record lightning-produced stokes as well as accurate time and location. Strokes





are then clustered into a flash if they are within 700 milliseconds and 10 kilometers. The detection coefficient is larger than 70% across southern contiguous US (Rudlosky, 2015).

For the comparison, the WRF-Chem simulation is that described in Sect. 2.2. ENTLN and WRF-Chem are sampled from May 13 to Jun 23 2012 over the middle and east US domain, where active lightning events are detected. Both observed and simulated lightning flashes are converted to flash density by dividing flash counts by corresponding grid areas and time range.

Figure 2a and b show the spatial distribution of flash density in number per $km^2$ per day observed by ENTLN and simulated by WRF-Chem. The largest biases are located over the southeast US (outlined by red on the map). In this region, WRF-Chem substantially overestimates flash density in general and a detection coefficient of 70% for ENTLN cannot account for the discrepancy. The simulated flash density is the highest primarily along the coast, which is not detected by ENTLN.

The scatter plot of daily flash density over the southeast US from two datasets in Fig. 2c demonstrates that the WRF-Chem consistently overestimates flashes in the southeast US over the study period. However, outside of the southeast US, the agreement improves. The simulation captures the spatial pattern over the regional scale (Fig. 2a–b) and the simulated flash densities are consistent with the observed flash densities and the correlation improves as well (Fig. 2d).

Currently, the cause of the discrepancies between the flash density from WRF-Chem simulation and ENTLN observation, is unknown. However, it is clear that it is the flash density, rather than the per-flash production rate of NO, is the cause of the disagreement in the UT between the daily profiles and SEAC4RS data. Further research is required to optimize the lightning parameterizations and improve flash density simulations in the southeast US for our model simulation.

## 3.2 Evaluation of variability in daily profiles

As demonstrated in Laughner et al. (2016), simulating the day-to-day variability in the a priori $NO_2$ profiles can have a significant impact on the retrieved $NO_2$ VCDs, due primarily to the day-to-day variation in wind speed and direction driving outflow from emissions sources, e.g. cities and power plants. To examine how well WRF-Chem captures the day-to-day variability in $NO_2$ profiles, we compare aircraft data from three DISCOVER-AQ campaigns and the matched WRF-Chem data (Sect. 2.4). For each profile in the DISCOVER data, we binned the $NO_2$ concentrations by pressure and calculated the correlation between WRF-Chem and aircraft $NO_2$ concentrations (one data point per profile per pressure bin). The results are shown in Fig. 3.

In California (Fig. 3a), the monthly average profiles correlate better with the aircraft data. However, as mentioned before, the Californian Central Valley is known to be difficult to model accurately (Hu et al., 2010). In Colorado (Fig. 3c), the daily profiles do a slightly better job overall, getting the variability at the surface and in an elevated layer more accurately than the monthly average profiles. The difference in Texas is quite dramatic (Fig. 3b), with the daily profiles performing substantially better. This suggests that daily profiles are able to capture variability caused by small, concentrated urban plumes much more effectively than monthly average profiles.

As a second check, we also compare WRF-Chem tropospheric VCDs to OMI SCDs to evaluate the general accuracy of wind direction and speed in the daily model profiles. The OMI SCDs do not depend on modeled vertical profiles, and so constitute an independent check on the plume direction. In order to have strong isolated $NO_x$ sources, we use Atlanta, Chicago, Las Vegas, Los Angeles, New York, and the Four Corners power plant for this study. For each of these sites, 5 days from 2007 are





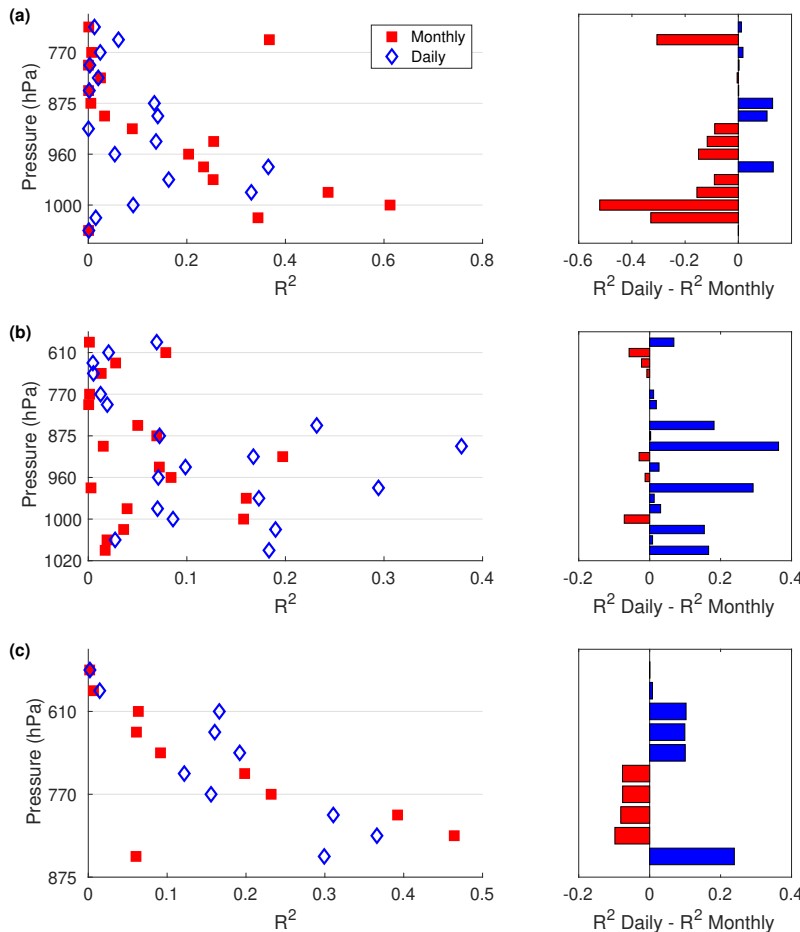

**Figure 3.** $R^2$ values for correlation between aircraft data and spatiotemporally matched WRF-Chem data for the **(a)** DISCOVER-CA, **(b)** DISCOVER-TX, and **(c)** DISCOVER-CO campaigns, binned by pressure. Left column: absolute $R^2$ values for each bin. Right column: the difference in $R^2$ values using monthly average and daily profiles for each bin.



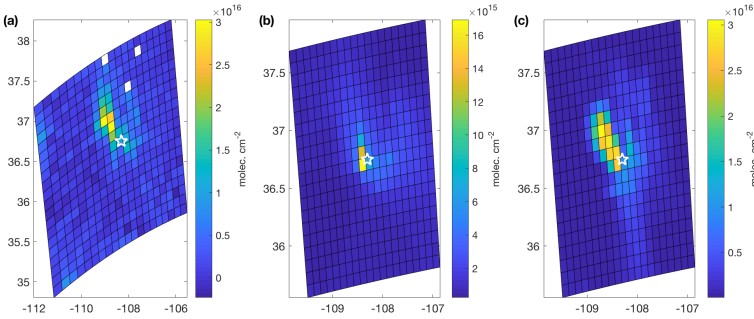

**Figure 4.** A comparison of OMI SCDs **(a)** and WRF monthly average **(b)** and daily **(c)** VCDs. The star marks the location of the Four Corners power plant. Data is from 4 Mar 2007.

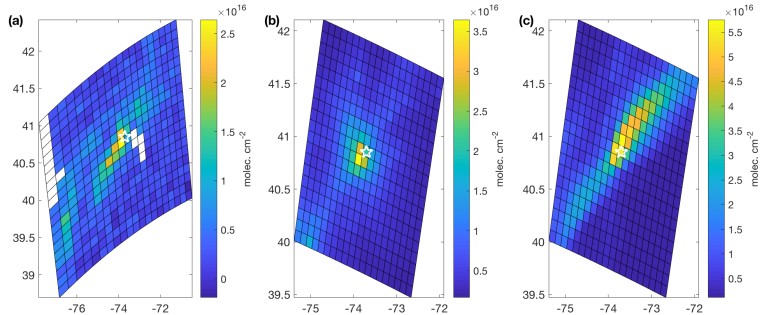

**Figure 5.** A comparison of OMI SCDs **(a)** and WRF monthly average **(b)** and daily **(c)** VCDs. The star marks the location of New York, NY, USA. Data is from 29 Sept 2007.

randomly chosen. If insufficient OMI SCDs are available for any day ($> 10\%$ of OMI pixels are cloud covered or in the row anomaly), another day is randomly chosen.

For each day, the agreement between the relative spatial distribution of WRF-Chem VCDs and OMI SCDs is manually evaluated, focusing on whether the model plume is advected in the same direction as the OMI SCDs indicate. Each day's agreement is evaluated qualitatively as good or bad. This, whether the WRF-Chem daily VCDs are significantly different from the monthly average WRF-Chem VCDs, and the confidence in the comparison are recorded for each comparison. Because of the number of factors that affect the absolute magnitude of SCDs, we look for qualitative, rather than statistically quantitative, agreement between the modeled VCDs and OMI SCDs. This is relevant since Laughner et al. (2016) noted that it is primarily the plume shape that drives the day-to-day variability in AMFs, therefore a direct, qualitative evaluation of the plume shape is desirable.

Figures 4 and 5 show two example comparisons, one good (Fig. 4) and one poor (Fig. 5). By studying randomly chosen days for 6 large $NO_x$ sources, we find that about 67–73% of days with sufficient data to be evaluated show good agreement between





the OMI SCDs and WRF-Chem daily VCDs. (The range is due to different levels of confidence filtering.) This indicates that the WRF-Chem simulated $NO_2$ profiles are adequately capturing the day-to-day variability due to wind speed and direction.

Both comparisons (vs. OMI SCDs and aircraft measurements) show that daily WRF-Chem profiles do, on average, a better job than monthly average profiles capturing the day-to-day variation in profile shape. Therefore, the core improvement in BEHR

v3.0, the transition to daily, high-resolution a priori profiles, is fundamentally sound. Daily profiles are especially important for applications that focus on upwind/downwind differences in $NO_2$ columns around a $NO_x$ source (Laughner et al., 2016) and, as we will see in Sect. 4, generally improve the retrieval in dense urban areas.

## 4   Column density evaluation

For the DISCOVER campaigns, we compare BEHR against aircraft-derived and Pandora VCDs together, calculating a single

regression line for the combined dataset. We use two comparisons, in one, only Pandora data that has a coincident aircraft profile is include ("matched"), in the other, all cloud-free Pandora data is used ("all"). Slopes and their $1\sigma$ uncertainties for combined aircraft and Pandora VCDs are shown in Table 2. For the DISCOVER-CO aircraft comparison, negative VCDs were removed. Such VCDs result from an overestimated stratosphere; since all versions of BEHR use the same stratosphere as their respective NASA SP products, an error in stratospheric subtraction will be present in all products, and since they cannot be

corrected in the BEHR retrieval, do not contribute useful information to the evaluation. To evaluate the southeast US, we use the SENEX and SEAC4RS campaigns, which only have aircraft data. These results are shown in Table 3. More details (slope, intercepts, $R^2$ values) can be found in Tables S1, S2, and S3.

In the following sections, we will evaluate the new BEHR v3.0 VCDs from three perspectives: performance compared to the current NASA SP, performance compared to the previous version of BEHR, and performance using daily a priori profiles

compared to using monthly a priori profiles.

### 4.1   Comparison vs. SP v3.0

For all the DISCOVER campaigns, BEHR v3.0 shows better agreement with both aircraft and Pandora measurements than the NASA SP v3.0 (slopes closer to 1). This is expected, since these campaigns generally centered on one or more cities, and a key feature of the BEHR retrieval are the $\sim 12$ km a priori profiles ($\sim 10x$ high resolution than the NASA SP v3.0 profiles) which

better capture the urban profile shape.

In the SENEX and SEAC4RS campaigns, BEHR's performance is more mixed. These campaigns include the southeast US, where we found that the WRF-Chem simulation that generated the a priori profiles overestimated the lightning flash density (Sect. 3.1). In SEAC4RS, whether BEHR v3.0 (M) performs better or worse than the NASA SP v3.0 depends on the method used to extend the profile (Sect. 2.4). This indicates that uncertainty in the measurement is greater than the difference between

these two products. BEHR v3.0 (D) performs poorly in the SENEX campaign; this will be explored in Sect. 4.3. Overall, BEHR v3.0 (M) is not significantly affected by the overestimated lightning flash density in the southeast US, as the monthly average profiles smooth out the overlarge UT lightning $NO_2$ signal.





| Campaign | Product | Slope (Matched) | Slope (All) |
|---|---|---|---|
| DISCOVER-MD | BEHR v3.0B (D) | N/A | N/A |
| | BEHR v3.0B (M) | $0.80 \pm 0.08$ | $0.64 \pm 0.03$ |
| | BEHR v2.1C | $1.3 \pm 0.1$ | $0.87 \pm 0.05$ |
| | SP v3.0 | $0.79 \pm 0.08$ | $0.50 \pm 0.03$ |
| DISCOVER-CA | BEHR v3.0B (D) | $0.49 \pm 0.04$ | $0.68 \pm 0.04$ |
| | BEHR v3.0B (M) | $0.51 \pm 0.04$ | $0.66 \pm 0.04$ |
| | BEHR v2.1C | $0.57 \pm 0.05$ | $0.68 \pm 0.04$ |
| | SP v3.0 | $0.41 \pm 0.04$ | $0.54 \pm 0.03$ |
| DISCOVER-TX | BEHR v3.0B (D) | $0.69 \pm 0.07$ | $1.00 \pm 0.06$ |
| | BEHR v3.0B (M) | $0.60 \pm 0.05$ | $0.87 \pm 0.05$ |
| | BEHR v2.1C | $1.1 \pm 0.1$ | $1.33 \pm 0.08$ |
| | SP v3.0 | $0.53 \pm 0.05$ | $0.74 \pm 0.05$ |
| DISCOVER-CO | BEHR v3.0B (D) | $0.66 \pm 0.06$ | $0.66 \pm 0.03$ |
| | BEHR v3.0B (M) | $0.70 \pm 0.06$ | $0.63 \pm 0.03$ |
| | BEHR v2.1C | $0.74 \pm 0.06$ | $0.68 \pm 0.03$ |
| | SP v3.0 | $0.53 \pm 0.05$ | $0.50 \pm 0.02$ |

**Table 2.** Slopes and $1\sigma$ uncertainties of BEHR vs. combined aircraft (extended with GEOS-Chem profiles) and Pandora VCDs. Matched slopes use only Pandora data approximately coincident with aircraft profiles to get similar sampling; all uses all valid Pandora data. Outliers and negative VCDs are removed before computing slopes.

| Campaign | Product | Slope (GEOS-Chem) | Slope (Extrap.) |
|---|---|---|---|
| SENEX | BEHR v3.0B (D) | $2.3 \pm 0.5$ | $1.7 \pm 0.5$ |
| | BEHR v3.0B (M) | $1.0 \pm 0.2$ | $0.9 \pm 0.3$ |
| | BEHR v2.1C | $1.4 \pm 0.4$ | $1.5 \pm 0.5$ |
| | SP v3.0 | $1.1 \pm 0.2$ | $0.8 \pm 0.3$ |
| SEAC4RS | BEHR v3.0B (D) | $0.9 \pm 0.4$ | $0.7 \pm 0.3$ |
| | BEHR v3.0B (M) | $1.2 \pm 0.4$ | $1.0 \pm 0.3$ |
| | BEHR v2.1C | $2.6 \pm 0.5$ | $2.5 \pm 0.7$ |
| | SP v3.0 | $1.0 \pm 0.3$ | $0.8 \pm 0.3$ |

**Table 3.** Slopes and $1\sigma$ uncertainties for RMA regression of satellite VCDs against in situ calculated VCDs. Both methods of extending the profiles (using GEOS-Chem modeled profiles or extrapolating the top/bottom ten points) are included. Outliers are removed before calculating these parameters.





### 4.2 Comparison vs. BEHR v2.1

Using just Pandora data coincident with aircraft spirals, v2.1 performs better in all DISCOVER campaigns except MD. However, using all Pandora data, v3.0 (D) performs better than or similar to v2.1 in all DISCOVER where daily profiles are available. The Pandoras provide more observations than the aircraft profiles, and, due to their small footprint, are more sensitive to narrow, highly concentrated $NO_2$ plumes. The v2.1 profiles used 2005 emissions; as seen in Fig. 1, this led to too much $NO_2$ being placed at the surface, which will increase the retrieved VCD. This suggests that the better performance of v2.1 in some cases is due to cancellation of errors; overestimated surface $NO_2$ is canceling out the lack of temporal variation in the profiles. That is, the higher average surface concentration in the v2.1 profiles may be similar to the in-plume concentrations resolved by the daily v3.0 profiles.

In v3.0, when daily profiles are available, the agreement is similar to or better than v2.1 if all Pandora data is used. Therefore, daily profiles are able to capture at least some enhancements in surface $NO_2$ where and when they occur, without overestimating the average profile. This is not evident using just the coincident Pandora data because of the smaller number of comparisons. As the comparison expands (using all Pandora data), the improvement becomes evident. The better performance of daily profiles suggests that even though Laughner et al. (2016) did not see large effects in a multi-month average using daily instead of monthly profiles, that daily profiles will provide a more accurate representation of urban VCDs over longer averaging periods.

BEHR v3.0 performs better in the SENEX and SEAC4RS comparisons than v2.1 (excluding 3.0 (D) in SENEX). The v2.1 profiles did not include lightning emissions, as it was a limitation of WRF-Chem at that time (Laughner et al., 2018f). This indicates that, even though the contribution of lightning to the southeast US profiles is too large, the inclusion of lightning $NO_2$ in the profiles did improve the representation of the southeast US. Laughner et al. (2018f) also showed that implementing a variable tropopause pressure decreased VCDs in the southeast US during summer; this also would help reduce the high bias compared to SENEX and SEAC4RS seen in BEHR v2.1.

### 4.3 Comparison of BEHR v3.0 (M) vs BEHR v3.0 (D) in the SE US

In the SENEX campaign, v3.0 (D) performs significantly worse than v3.0 (M). From Fig. 1 we know that the daily a priori profiles overestimate the UT $NO_2$, and from Fig. 2 we know that this is due to a significant overestimate of the flash density in our WRF simulation. The comparison in Table 3 would seem to indicate that this overestimate has a severe impact on the retrieved VCDs, but we must also consider the uncertainty in the SENEX-derived VCDs.

Figure 6a shows the ensemble of profiles from SENEX used to calculate VCDs. The circles mark levels that had to be calculated using model data for $> 50\%$ of the profiles. In SENEX, that is all levels about $\sim 700$ hPa, which means that the SENEX aircraft data provides very little constraint on the UT. The lightning contribution to the SENEX columns must come from the GEOS-Chem monthly averages or extrapolation from a lower altitude, which means the spatial and temporal variation is lost.

Figure 6c shows the effect of using the WRF-Chem a priori profiles instead of the GEOS-Chem profiles to extend the SENEX profiles. The WRF-Chem profiles do include spatial and temporal variation of the UT, but using them reinforces the





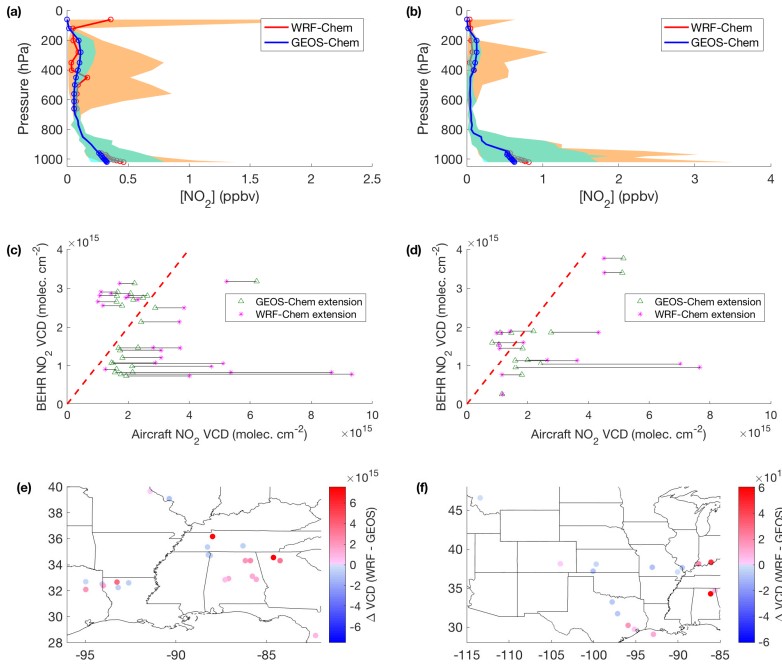

**Figure 6. (a,b)** The profiles used to calculate the aircraft VCDs extended using WRF-Chem or GEOS-Chem profiles; the solid line is the median of all profiles, the shading represents the 10th and 90th percentiles for each binned level. Circles indicate levels that were derived from the models in at least 50% of the profiles. **(c,d)** Comparison of BEHR v3.0 (D) VCDs vs. aircraft-derived VCDs using GEOS-Chem and WRF-Chem profiles to extend the profile to the surface and tropopause. The black lines connect corresponding comparisons between the two methods and the red dashed line represents the 1:1 agreement. **(e,f)** Difference between aircraft VCDs extended with WRF-Chem and GEOS-Chem profiles. **(a,c,e)** are for the SENEX campaign, **(b,d,f)** are for SEAC4RS.

AMF errors, moving all points away from the 1:1 line. Without either in situ measurements of the UT in the southeast US or Pandora total column observations we cannot separate the errors in AMF caused by the overestimated UT $NO_2$ in the a priori profiles from the error caused by the lack of spatiotemporal variation in the extended aircraft profiles. For example, the error in the cluster of points below the 1:1 line in Fig. 6c could be corrected if either the UT $NO_2$ in the a priori profile was reduced, decreasing the AMFs and so increasing the BEHR VCDs, or if the aircraft profile had less $NO_2$, thus moving the points left onto the 1:1 line. (In the case, there would still be a discrepancy between the BEHR VCD and the VCD derived from combining aircraft and WRF-Chem profiles, suggesting that the WRF-Chem UT $NO_2$ is still too great.)

Other campaigns do have better sampling of the UT, e.g. SEAC4RS (Fig. 6b,d,f), but do not have as many profiles in the southeast US (Fig. 6e,f). Therefore, we must currently assign an uncertainty of $\pm100\%$ to VCDs retrieved with daily profiles in the southeast US (east of $95°$ W and south of $37.5°$ N). This is almost certainly overly conservative, as Laughner et al. (2018f)





showed that the frequency distribution of UT $NO_2$ in the southeast apriori profiles was skewed to lower values in the daily profiles, and a three month average using daily a priori profiles resulted in greater VCDs than using monthly a priori profiles, which would not be the case if the daily profiles always overestimated the UT. This suggests that days with little or no lightning in both the real world and WRF-Chem simulations are more numerous than days with significant lightning contribution, and

so a multi-month average using daily profiles would in fact accurately capture this. However, without long term independent column measurements in the southeast, we cannot confirm this hypothesis. Future work will focus on improving the simulation of lightning in the southeast US. If successful, improved WRF-Chem profiles for the southeast can be implemented.

## 4.4  Comparison of BEHR v3.0 (M) vs BEHR v3.0 (D) in urban areas

In the DISCOVER campaigns, BEHR v3.0 (D) using daily profiles has regression slopes similar to or closer to 1 than

BEHR v3.0 (M) using monthly profiles except in the DISCOVER-CA aircraft comparisons. There is a clear improvement in DISCOVER-TX using daily profiles. This suggests that the daily profiles are capturing small, concentrated plumes in the urban area (Fig. 3b), which is improving the retrieval overall in an urban area with many highly concentration industrial $NO_x$ sources. Therefore, we argue that daily profiles improve the retrieval in many ways, not only for applications that select for upwind/downwind pixels as shown in Laughner et al. (2016), but also for multi-month averages in dense urban areas.

## 5  Discussion: future efforts to validate daily profiles

Using space-based SCDs to evaluate the spatial distribution of $NO_2$ in a CTM is powerful (Sect. 3.2), because both provide a spatially continuous field of $NO_2$ columns. As we have shown here, this makes a qualitative evaluation straightforward and illustrative. However, a quantitative metric is more challenging to devise, as the direct correlation of model and satellite columns is less important than the more abstract agreement between the overall plume direction and extent. As we have shown here, daily, high-resolution profiles provide important benefits to an $NO_2$ retrieval; therefore, development of more quantitative

methods to evaluate model performance in this manner should be a priority.

There are several possibilities. First, an algorithm that identifies the plume and computes the direction and length of its major axis could be used. This would allow a comparison of the direction and extent of the plumes more directly. Such an algorithm would not be trivial to develop; comparisons such as the one shown in Fig. 5a,c would likely be difficult for the algorithm to

distinguish the plume direction accurately.

Second, this problem could be treated as an image recognition problem. A neural network could be trained on modeled VCDs and SCDs. A training set of good and bad days could be constructed from the WRF-Chem simulations used in BEHR v3.0D. Development of this approach is beyond the scope of this paper.

Third, dense sensor networks (e.g. Shusterman et al., 2016; Kim et al., 2018) may also be useful to evalulate daily profiles by

permitting a simpler correlation test between modeled and observed surface concentrations than is possible between modeled VCDs and observed SCDs. Development of these networks is a topic of active research. This method may be necessary for





future retrievals, especially over the US and European domains, where decreasing $NO_x$ emissions mean that the contrast between plumes and background in SCDs is much weaker now than in 2007.

## 6   Conclusions

We have evaluated version 3.0B of the BEHR OMI $NO_2$ product against multiple datasets. We find that the WRF simulation
used to generate the a priori $NO_2$ profiles generally agrees well with the available aircraft data; however, the number of lightning flashes is significantly overestimated in the southeast US leading to an overestimate of the UT $NO_2$ in that region, although broadly consistent with ENTLN observations elsewhere. When compared against aircraft-derived and Pandora VCDs, BEHR v3.0B performs better than SP v3.0, with regionally varying low biases of 0–51% compared to in situ and Pandora measurements. Using daily profiles yields better results than monthly profiles, except in the southeast US.

The lessons learned here are applicable to geostationary satellites scheduled to launch in the near future. Because the BEHR retrieval focuses on the continental United States, it serves as a useful prototype for future $NO_2$ retrievals from geostationary satellites such as GEMS (Bak et al., 2013; Choi and Ho, 2015), Sentinel-5 (Ingmann et al., 2012), and TEMPO (Chance et al., 2013), which also will be inherently restricted to regional areas. This offers the opportunity to use higher-resolution a priori data than global retrievals.

Here, the results from the SENEX and SEAC4RS campaign here demonstrate that verifying the chemical transport model's reproduction of the day-to-day variability in lightning flashes is vital to obtain reliable results in such regions. With the sub-daily temporal resolution available to geostationary satellites, this will only become more important. Therefore, geostationary retrievals should evaluate the diurnal variation in lightning flashes in their a priori models using ground- and space- based lightning detectors (e.g. NLDN, ENTLN, or the GOES-R lightning mapper), and plans should be made to validate retrieved
VCDs in multiple regions that have strong, but different, lightning influence. Such validations must include measurement of the UT $NO_2$ profile and/or total column observations in order to reliably separate errors in the a priori profiles from errors in the observations used for evaluation.

Evaluating the day-to-day performance of the a priori profiles in future geostationary retrievals is crucial. Daily profiles have been shown to significantly affect retrieved $NO_2$, especially in applications that systematically focus on $NO_2$ VCDs downwind
of a source (Laughner et al., 2016), and we have shown here that daily profiles also improve performance in urban areas. With the WRF-Chem model configuration used here, urban $NO_2$ plumes are simulated with the correct spatial pattern $\sim 70\%$ of the time. Planned campaigns to evaluate geostationary satellite retrievals should be designed with an eye towards also evaluating the day-to-day accuracy of the a priori profiles.

*Code and data availability.*   The analysis code for this paper is available at https://github.com/behr-github/BEHR-v3-evaluation/ (Laughner,
2018). Supporting datasets generated or used by this code is hosted by UC Dash (Laughner et al., 2018e). The BEHR v3.0B product is hosted




as four subproducts by UC Dash (Laughner et al., 2018a, b, c, d) as well as on behr.cchem.berkeley.edu. The BEHR algorithm is available at https://github.com/CohenBerkeleyLab/BEHR-core/tree/master (Laughner and Zhu, 2018).

*Competing interests.*   The authors declare no competing interests.

*Acknowledgements.*   The authors gratefully acknowledge support from the NASA ESS Fellowship NNX14AK89H, NASA grant NNX15AE37G,

5   and the TEMPO project grant SV3-83019.

We would like to acknowledge high-performance computing support from Cheyenne (doi:10.5065/D6RX99HX) provided by NCAR's Computational and Information Systems Laboratory, sponsored by the National Science Foundation. This research also used the Savio computational cluster resource provided by the Berkeley Research Computing program at the University of California, Berkeley (supported by the UC Berkeley Chancellor, Vice Chancellor for Research, and Chief Information Officer).

10   We acknowledge use of the WRF-Chem preprocessor tools MOZBC, fire_emiss, etc. provided by the Atmospheric Chemistry Observations and Modeling (ACOM) laboratory of NCAR.

Finally, we acknowledge the use of in situ $NO_2$ measurements from the UC Berkeley TD-LIF instrument and the NOAA chemiluminescence instrument, as well as ground based $NO_2$ columns from multiple Pandora spectrometers operated by various groups throughout the DISCOVER-AQ campaigns.





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
