# Peer review of "Evaluation of version 3.0B of the BEHR OMI NO2 product"

_Atmospheric Measurement Techniques, 2018_

## Referee Comment (RC1) · Anonymous Referee #1 · 19 Sep 2018

In this manuscript, the authors evaluate the performance of their updated tropospheric NO2 vertical column product (and the model output needed to derive it) against ground-based vertical column observations and in-situ aircraft profiles. They find that, in general, the updated retrieval is an improvement over the previous version and over the latest NASA standard product. The paper demonstrates the improvement in retrieval performance that is achieved by using daily a-priori NO2 profiles over monthly averages. This paper also makes a number of pertinent key arguments that will be of interest to the wider community, e.g.: (1) that there is a lack of measurements in the southeast US to constrain sources of uncertainty in the retrieval of NO2 in that region; (2) the need for further development in dense sensor networks that can evaluate contrast between pollutant plumes and background concentrations; and (3) the value of

using slant column densities to evaluate the spatial distribution of NO2 predicted by a model.

Overall, this manuscript is detailed and clearly written. It covers subject matter that is certainly relevant to the AMT audience, and should be of interest to the community. I would recommend this paper for publication in AMT, and make some small recommendations that could be considered prior to acceptance.

General Comment:

This manuscript seems to focus largely on how the updated model chemistry/emissions and the daily vs. monthly averaged a-priori profiles impact the performance of the BEHR NO2 retrieval. But there were other pertinent updates in the retrieval: varying tropopause height, directional surface reflectance, and a new combined surface pressure dataset. Are the differences in retrieval performance against observations as a result of these updates very minor compared to the differences attributed to the updated model chemistry and profile temporal resolution? It seems worthwhile mentioning/commenting on. I wondered whether there were any specific instances/cases where some of these other updates could be relatively more important.

Detailed comments:

1) Abstract: I suggest including the detail that BEHR is focused on retrievals over North America in the abstract.

2) Introduction: The introduction includes some background on previous evaluation efforts for the NASA SP and KNMI DOMINO products. Given the focus of this paper on evaluating this latest version of the BEHR algorithm, some details on how previous versions of this algorithm have been evaluated (and its performance) also seems relevant. This could add more motivation/context for the necessity of updating the algorithm (in addition to the already cited work from Laughner et al. 2016 that focused on the importance of daily profiles).

3) p. 5, l. 29-30: The authors mention using the GEOS-Chem global chemistry model to extend aircraft profiles to the surface. It could be relevant here to include at least the horizontal resolution of the model output used. Is the model output identical to the model experiment run cited in Nault et al. (2017)? It wasn't clear to me whether these authors were directly using output from that experiment.

4) p. 6, l. 18: I wonder whether some summary statistics might more easily advance the authors' argument that model output from v3.0 "show better agreement" than model output from v2.1. At the moment, there is only a description of the more obvious qualitative details.

5) p. 6, l. 31: "since the strongest lighting occurs. . ." Replace "lighting" with "lightning"

6) p. 9, l. 29: I suggest inserting "modeled" between "daily" and "profiles" so the point is very clear.

7) p. 12, l. 2: The word "adequately" is used rather subjectively here. I know it is discussed later in the manuscript, but I wonder whether there is any way to quantitatively evaluate just how well the WRF-Chem NO2 profiles are indeed capturing the day-to-day variability. If not, perhaps this caveat could be more clear, and you could mention here that you propose some suggestions later in the manuscript.

8) p. 12, l. 12-15: This explanation of negative VCDs in the DISCOVER-CO dataset is unclear to me.

9) p. 12, l. 28-30: The authors introduce the notation of "BEHR v.30 (M)" and "(D)" for the first time here, I think. While it's fair to say it is obvious, the authors could explicitly clarify that "(M)" refers to the product using monthly average profiles, and "(D)" refers to the product using daily profiles.

10) Table 2 and 3: I was wondering whether there would be value in reporting correlation coefficients in addition to slopes. Can the authors explain why they haven't included these in their evaluation of the product performance? For example, I wondered whether they can demonstrate that in addition to improving a bias, using daily profiles (D) explain more of the variability than monthly profiles (M) alone.

11) Section 4.2: Can the authors clarify why they have chosen to separate their evaluation by looking at all the Pandora data, vs. just the Pandora data during coincident aircraft spirals? Is this meant to demonstrate how continued long-term monitoring is superior to short-term campaign coverage for evaluation purposes?

---

## Referee Comment (RC2) · Anonymous Referee #2 · 23 Oct 2018

General Comments

The manuscript presents the comparison of an improved version of BEHR OMI NO2 product with NO2 vertical profile and total columns. The results show that the new BEHR NO2 product V3.0B performs better than previous versions and than OMI NO2 standard product, as compared to aircraft and Pandora ground-based measurements. The authors recommend the use of daily NO2 profiles when designing new algorithms fo geostationary missions. They also remind the importance of NO2 production from lightning in south-east US, that should be corrected. The manuscript is generally sound but it is quite technical, due to the large amount of different dataset compared to each other. I would suggest a few more plots for clarity. I recommend publication after addressing the following comments.

[Figure]

Specific Comments

1. Section 4. The comparison of VCD should include some scatterplots. Presenting the comparison as a table (Table 2) only is a bit difficult to follow. Especially, the results with separated Pandora and aircraft VCDs should be shown. Some of these plots could go in the supplementary material.

2. You use a quite precise spatial collocation criteria (pandora site within the OMI pixel, so no spatial smoothing in practice) and then you time average Pandora observations ±1 h from the OMI observation, which is quite large time frame. Can you open a little bit about this choice? How does the results change with a shorter time interval?

3. You mention several times in the text that changes in emission information as input have a role in the discrepancies you observe between different versions of the algorithm. Could you actually show them? For example, plotting or mentioning the quantitative the emission changes over the areas of study.

Technical comments

4. Fig. 1: the plot is quite small, it could be a bit bigger so that the the different lines can be better separated? Also, the legend can be shown only once.

5. P12 L11 only Pandora data that has a coincident aircraft profile is include -> only Pandora data that have a coincident aircraft profile are included

6. Conclusions P17 L12 I think you refer to Sentinel 4, as Sentinel 5 is not planned on a geostationary orbit.

7. Figure 1 and 3: the different panels would benefit from a title mentioning the different sites considered in the plot

––––––––––––––––––––––––

---

## Author Comment (AC1) · 9 Nov 2018

**Evaluation of version 3.0B of the BEHR OMI NO$_2$ product**

**Response to Anonymous Referee #1**

Joshua L. Laughner, Qindan Zhu, and Ronald C. Cohen

November 7, 2018

We thank the reviewer for their positive response and careful reading. The suggestions offered and questions raised are well taken, and we have done our best to incorporate them into the paper.

Responses to specific comments follow. The reviewer's comments will be shown in red, our response in blue, and changes made to the paper are shown in black block quotes. Unless otherwise indicated, page and line numbers correspond to the original paper. Figures, tables, or equations referenced as "R$n$" are numbered within this response; if these are used in the changes to the paper, they will be replaced with the proper number in the final paper. Figures, tables, and equations numbered normally refer to the numbers in the original discussion paper.

This manuscript seems to focus largely on how the updated model chemistry/emissions and the daily vs. monthly averaged a-priori profiles impact the performance of the BEHR NO2 retrieval. But there were other pertinent updates in the retrieval: vary- ing tropopause height, directional surface reflectance, and a new combined surface pressure dataset. Are the differences in retrieval performance against observations as a result of these updates very minor compared to the differences attributed to the updated model chemistry and profile temporal resolution? It seems worthwhile men- tioning/commenting on. I wondered whether there were any specific instances/cases where some of these other updates could be relatively more important.

Our decision to focus most strongly on the modeled NO$_2$ profiles was due primarily to two reasons:

1. In discussions with other members of the NO$_2$ remote sensing community, concerns were raised about the accuracy of such high spatial and temporal resolution profiles.

2. In Laughner et al. (2018), we showed that the effect of the a priori profiles on the NO$_2$ VCDs has one of the two greatest effects on the NO$_2$ VCDs.

Because accurately simulating day-by-day high resolution NO$_2$ profiles accurately is challenging, and because to our knowledge, this is the first time an NO$_2$ product using such profiles has been generated for such a large time period and domain, we felt it was most important to validate that component separately. The other scientifically interesting modifications either are or will likely be standard for state-of-the-art NO$_2$ retrievals (BRDF,

Vasilkov et al. 2017; surface pressure, Zhou et al. 2009; variable tropopause, Bucsela et al. 2013).

During testing, we had found that using the hypsometric surface pressure correction recommended by Zhou et al. (2009) (instead of the simple scale height relationship used in v3.0A and earlier) had a comparatively large effect on the agreement between BEHR and aircraft + Pandora VCDs during the DISCOVER-CO campaign, which took place in the Rocky Mountains. This is interesting, because the Zhou et al. (2009) method was originally intended to downscale very coarse resolution ($3° \times 2°$) modeled surface pressure to OMI pixel resolution; we did not expect it to provide a large advantage over using an already high resolution surface elevation database with a scale height calculation. We have added a short section describing this to the supplement:

> "While we did not carry out an explicit test of how each change to the BEHR algorithm between v2.1C and v3.0B affected the comparison vs. aircraft and Pandora data, we did investigate the effect of different methods of computing the surface pressure of the OMI pixels. The AMF calculation requires a priori knowledge of the average surface pressure of the each OMI pixel, as the location of the surface affects the shape of the scattering weights (e.g. a low reflectivity surface high up in the atmosphere will cause the scattering weights to decrease more rapidly with decreasing altitude than a surface lower down in the atmosphere).
>
> In BEHR v3.0A and earlier versions, this surface pressure was computed by averaging surface elevation data from the GLOBE database (Hastings and Dunbar, 1999) within the OMI pixel, which is then converted to from elevation to pressure using a 7.4 km scale height. In v3.0B, surface pressure taken from the same WRF-Chem model that supplied the $NO_2$ profiles is adjusted using the same average GLOBE surface elevation in the method described by Zhou et al. (2009). The Zhou et al. (2009) method was originally intended to downscale very coarse ($\sim 3° \times 2°$) modeled surface pressure to OMI pixels using a high resolution terrain database, therefore the effect of using it with already high resolution modeled surface pressure has not been tested.
>
> Laughner et al. (2018) showed that switching to the (Zhou et al., 2009) method increased BEHR $NO_2$ VCDs by $\sim 5$ to $10\%$ over the Rocky Mountains during the summer months. This is a small but systematic change, and so was investigated as a way to correct the low bias in BEHR $NO_2$ VCDs vs. aircraft and Pandora measurements during the DISCOVER-AQ Colorado campaign.
>
> Figure R1 shows regressions of BEHR VCDs against aircraft + Pandora VCDs using both the scale height and hypsometric equation methods of computing surface pressure. The latter method improves the slope by $\sim 18\%$, and while there is a small increase in most BEHR VCDs, the reduction of the 4 largest aircraft VCDs has a larger effect on the slope. The aircraft VCDs change because, when computing a VCD from the aircraft profiles, we integrate from the OMI pixel surface pressure to its tropopause pressure, for consistency between the aircraft VCD calculation and BEHR AMF calculation.

[Figure]

Figure R1: Regression of BEHR v3.0B (D) VCDs vs. aircraft + matched Pandora VCDs for the DISCOVER-CO campaign. The blue series and fit used surface pressured computed using a 7.4 km scale height to convert GLOBE elevations into pressures; the red series used the hypsometric equation as in Zhou et al. (2009).

As the two methods of calculating surface pressure do not significantly alter the BEHR VCDs in this comparison, we cannot say explicitly that the surface pressure calculated with the hypsometric equation improves the BEHR retrieval. However, using that surface pressure does lead to greater consistency between BEHR and aircraft VCDs when also applied as the lower limit for integrating the aircraft profiles."

Abstract: I suggest including the detail that BEHR is focused on retrievals over North America in the abstract.
We have added:

"Version 3.0B of the Berkeley High Resolution (BEHR) OMI $NO_2$ product is designed to accurately retrieve daily variation in the high spatial resolution mapping of tropospheric column $NO_2$ **over continental North America between 25° N and 50° N.**"

Introduction: The introduction includes some background on previous evaluation ef- forts for the NASA SP and KNMI DOMINO products. Given the focus of this paper on evaluating this latest version of the BEHR algorithm, some details on how previ- ous versions of this algorithm have been evaluated (and its performance) also seems relevant. This could add more motivation/context for the necessity of updating the al- gorithm (in addition to the

already cited work from Laughner et al. 2016 that focused on the importance of daily profiles).

We have added the following paragraph near the end of the introduction:

"Russell et al. (2011) evaluated the original BEHR algorithm over California using data from the Arctic Research of the Composition of the Troposphere from Aircraft and Satellites (ARCTAS-CA) field campaign. As the ARCTAS-CA campaign did not include a large number of tropospheric profiles, Russell et al. (2011) computed aircraft-derived $NO_2$ VCDs from times when the aircraft was flying in the boundary layer. Assuming a well-mixed boundary layer, Russell et al. (2011) extrapolated the measurements within the boundary layer to the surface, and combined with measurements in the free troposphere from the remainder of the ARCTAS-CA campaign, were able to estimate tropospheric $NO_2$ VCDs from aircraft measurements for a larger number of coincident OMI pixels than would have been possible with traditional aircraft profiles, at the expense of increased uncertainty in the aircraft-derived VCDs. Russell et al. (2011) found that both the original BEHR product had similar agreement as the NASA SP v1 product with the aircraft data (both with slopes near 1), but BEHR had better correlation ($R^2$ 0.83 vs. 0.72). Since then, the plethora of aircraft campaigns and expansion of the Pandora ground based spectrometer network across the United States has provided better datasets to evaluate the BEHR product in a variety of locations."

p. 5, l. 29-30: The authors mention using the GEOS-Chem global chemistry model to extend aircraft profiles to the surface. It could be relevant here to include at least the horizontal resolution of the model output used. Is the model output identical to the model experiment run cited in Nault et al. (2017)? It wasn't clear to me whether these authors were directly using output from that experiment.

This is data from the same GEOS-Chem simulation as Nault et al. 2017. We have clarified this:

"We use modeled $NO_2$ profiles from the "updated + 33%" GEOS-Chem simulation described in Nault et al. (2017) (v9.02 of the GEOS-Chem global chemical transport model (Bey et al., 2001) at $2.5° \times 2°$ resolution, with updated $HNO_3$, $HO_2NO_2$, and $N_2O_5$ chemistry and lightning emission rates)."

p. 6, l. 18: I wonder whether some summary statistics might more easily advance the authors' argument that model output from v3.0 "show better agreement" than model output from v2.1. At the moment, there is only a description of the more obvious qualitative details.

We have added Table R1 and the following paragraph after the second paragraph of section 3.1:

"

We evaluate the agreement quantitatively by calculating the mean absolute bias between the average WRF and aircraft profiles (Table R1). We divide the profiles

| | BL ($p > 775$ hPa) | BL (no SEAC4RS) | FT ($p \leq 775$ hPa) | FT (no SEAC4RS) |
|---|---|---|---|---|
| V2 | 965 | 902 | 71 | 86 |
| V3 Monthly | 530 | 609 | 74 | 87 |
| V3 Daily | 482 | 618 | 108 | 66 |

Table R1: Mean absolute bias between each of the types of simulated $NO_2$ profiles and the aircraft profiles shown in Fig. 1. Values are given for the boundary layer (BL) and free troposphere (FT), with the divide at 775 hPa ($\sim 2$ km). All values are in parts per trillion by volume (pptv).

> into boundary layer (BL) and free troposphere (FT), as different processes (e.g. anthropogenic vs. lightning emissions) govern them. As the SEAC4RS campaign has an obvious error in the free troposphere (which will be discussed below), we calculate these values with and without the SEAC4RS campaign. In the BL, the version 3 profiles have one-half to two-thirds the bias of the version 2 profiles (depending if SEAC4RS is excluded). In the free troposphere, there is little difference in the mean bias between profile types, unless SEAC4RS is included, in which case the daily profiles have a 33% greater bias."

p. 6, l. 31: "since the strongest lighting occurs. . ." Replace "lighting" with "lightning"
Corrected, thank you.

p. 9, l. 29: I suggest inserting "modeled" between "daily" and "profiles" so the point is very clear.
Added.

p. 12, l. 2: The word "adequately" is used rather subjectively here. I know it is discussed later in the manuscript, but I wonder whether there is any way to quantitatively evaluate just how well the WRF-Chem NO2 profiles are indeed capturing the day-to-day variability. If not, perhaps this caveat could be more clear, and you could mention here that you propose some suggestions later in the manuscript.
Unfortunately, developing a quantitative metric that accurately evaluates the important aspects of the agreement between the daily variability in the modeled $NO_2$ profiles and the real world is quite difficult. Structure functions have been used with success before (Follette-Cook et al., 2015), but are insufficient for our application: since structure functions measure average difference vs. separation, a simulated plume and a real plume with similar shapes but going in different directions will still have similar structure functions, even though they differ in a crucial way.
Developing the necessary methods to quantify how well the simulated daily profiles match reality would be an interesting and worthwhile endeavor, unfortunately it is outside the scope of this paper. Therefore we have added the following text to Sect. 3.2:

> "While we recognize that this conclusion is highly qualitative, the specific character of agreement that is important for these profiles (overall plume size and direction, rather than exact agreement between modeled and real concentrations

or column densities) is rather difficult to evaluate quantitatively. We recognize that developing such methods is necessary and offer several possible approaches in Sect. 5."

p. 12, l. 12-15: This explanation of negative VCDs in the DISCOVER-CO dataset is unclear to me.

We have expanded this explanation and reorganized the surrounding text slightly to accomodate it:

"For the DISCOVER-CO aircraft comparison, negative VCDs were removed. Negative VCDs occur when the estimated stratospheric $NO_2$ column is greater than the total $NO_2$ column, thus $V_{trop} = V_{total} - V_{strat} < 0$; they cannot be introduced by the AMF correction of the tropospheric SCD to VCD as the AMF is a multiplicative factor and always $> 0$. Since all versions of BEHR use the same stratospheric $NO_2$ column as their respective NASA SP products, an error in stratospheric subtraction will be present in all products, and it cannot be corrected in the BEHR retrieval. Aircraft VCDs, by their nature, cannot be negative, so for these comparisons we remove the negative VCDs so as to avoid increasing the regression slopes by trying to fit these erroneous points. (However, we do note that this is a special case where individual pixels or small groups of pixels are being compared against other VCDs. Most applications of BEHR data should retain the negative VCDs to avoid transforming the essentially Gaussian random stratospheric error into a systematic error by removing part of the bell curve.) Since the stratospheric VCDs are added back to the BEHR or NASA SP tropospheric VCDs for comparison with the Pandora VCDs, negative VCDs are not an issue with Pandora comparisons."

p. 12, l. 28-30: The authors introduce the notation of "BEHR v.30 (M)" and "(D)" for the first time here, I think. While it's fair to say it is obvious, the authors could explicitly clarify that "(M)" refers to the product using monthly average profiles, and "(D)" refers to the product using daily profiles

We have added the following sentence to the end of Sect. 4:

"Throughout, BEHR v3.0 (M) refers to BEHR using monthly $NO_2$ profiles; likewise, BEHR v3.0 (D) refers to the product using daily $NO_2$ profiles."

Table 2 and 3: I was wondering whether there would be value in reporting correlation coefficients in addition to slopes. Can the authors explain why they haven't included these in their evaluation of the product performance? For example, I wondered whether they can demonstrate that in addition to improving a bias, using daily profiles (D) explain more of the variability than monthly profiles (M) alone.

We do report $R^2$ values in the supplement (Table S3). We chose not to include them in the main paper as this section was already fairly dense (with a three-way comparison between

NASA, BEHR v2, and BEHR v3) and the slopes are the main indicator of the accuracy of each products' columns. Further, there is no clear pattern in the $R^2$ values. We have added the following sentence in Sect. 4 stating this:

> "We will focus on the regression slopes here; intercepts and $R^2$ values are given in Table S3 in the supplement; however we note that there is not a clear pattern of any one product having a consistently better $R^2$ value than the others."

 Can the authors clarify why they have chosen to separate their evaluation by looking at all the Pandora data, vs. just the Pandora data during coincident aircraft spirals? Is this meant to demonstrate how continued long-term monitoring is superior to short-term campaign coverage for evaluation purposes?

We should clarify that both cases include aircraft and Pandora data. Our main purpose was to provide both a regression that used all available data and one that weights the Pandora and aircraft data equally; when using all Pandora data, the sheer number of data points essentially overwhelms the aircraft data. Since both measurements have strengths and weaknesses for comparison with satellite columns, we wanted to show both a comparison over long time periods (all Pandora data) and one that gave the aircraft data fair weight. We have expanded Sect. 4 to describe this:

> "For the DISCOVER campaigns, we compare BEHR against aircraft-derived and Pandora VCDs together, calculating a single regression line for the combined dataset. These two measurements have unique strengths and weaknesses for comparison against satellite VCDs: Pandoras give a precise column measurement and can be deployed for long time periods, but have a very small footprint (leading to possible representativeness errors) and provide a total, not tropospheric, column. Aircraft profiles have a footprint more similar to an OMI pixel size, but introduce uncertainty due to missing parts of the profile (near the surface and in the upper troposphere in the DISCOVER campaigns) and cannot be deployed for long term, routine observations.
>
> In order to take advantage of each methods' strenghts, we use two comparisons, in one, only Pandora data that has a coincident aircraft profile is include ("matched"), in the other, all cloud-free Pandora data is used ("all"). We do so because, when including all Pandora data, the number of Pandora comparisons available will overwhelm the number of available aircraft profiles in the regression. Therefore the regressions using all Pandora data are representative of longer time periods, but weighted strongly towards the Pandora data, and the regressions using only the coincident data represent shorter time periods, but give more weight to the aircraft data."

We also remind the reader that these regressions are aircraft+Pandora in Sect. 4.2:

> "Using **aircraft data plus** just Pandora data coincident with aircraft spirals, v2.1 performs better.... However, using **aircraft data plus** all Pandora data, v3.0 (D) performs better...."

**References**

Bey, I., Jacob, D. J., Yantosca, R. M., Logan, J. A., Field, B. D., Fiore, A. M., Li, Q., Liu, H. Y., Mickley, L. J., and Schultz., M. G.: Global modeling of tropospheric chemistry with assimilated meteorology, J. Geophys. Res., 106, 23 073–23 096, 2001.

Bucsela, E., Krotkov, N., Celarier, E., Lamsal, L., Swartz, W., Bhartia, P., Boersma, K., Veefkind, J., Gleason, J., and Pickering, K.: "A new tropospheric and stratospheric $NO_2$ retrieval algorithm for nadir-viewing satellite instruments: applications to OMI, Atmos. Meas. Tech., 6, 2607–2626, doi:10.5194/amt-6-2607-2013, 2013.

Follette-Cook, M. B., Pickering, K. E., Crawford, J. H., Duncan, B. N., Loughner, C. P., Diskin, G. S., Fried, A., and Weinheimer, A. J.: Spatial and temporal variability of trace gas columns derived from WRF/Chem regional model output: Planning for geostationary observations of atmospheric composition, Atmos. Environ., 118, 28–44, doi:10.1016/j.atmosenv.2015.07.024, URL https://doi.org/10.1016/j.atmosenv.2015.07.024, 2015.

Hastings, D. and Dunbar, P.: Global Land One-kilometer Base Elevation (GLOBE) Digital Elevation Model, Documentation, Volume 1.0. National Oceanic and Atmospheric Administration, National Geophysical Data Center, 325 Broadway, Boulder, Colorado 80303, U.S.A., 1999.

Laughner, J. L., Zhu, Q., and Cohen, R. C.: The Berkeley High Resolution Tropospheric $NO_2$ Product, Earth System Science Data Discussions, 2018, 1–33, doi:10.5194/essd-2018-66, URL https://www.earth-syst-sci-data-discuss.net/essd-2018-66/, 2018.

Nault, B. A., Laughner, J. L., Wooldridge, P. J., Crounse, J. D., Dibb, J., Diskin, G., Peischl, J., Podolske, J. R., Pollack, I. B., Ryerson, T. B., Scheuer, E., Wennberg, P. O., and Cohen, R. C.: Lightning $NO_x$ Emissions: Reconciling Measured and Modeled Estimates With Updated $NO_x$ Chemistry, Geophys. Res. Lett., doi:10.1002/2017GL074436, 2017.

Russell, A. R., Perring, A. E., Valin, L. C., Bucsela, E. J., Browne, E. C., Wooldridge, P. J., and Cohen, R. C.: A high spatial resolution retrieval of $NO_2$ column densities from OMI: method and evaluation, Atmos. Chem. Phys., 11, 8543–8554, doi:10.5194/acp-11-8543-2011, URL https://doi.org/10.5194/acp-11-8543-2011, 2011.

Vasilkov, A., Qin, W., Krotkov, N., Lamsal, L., Spurr, R., Haffner, D., Joiner, J., Yang, E.-S., and Marchenko, S.: Accounting for the effects of surface BRDF on satellite cloud and trace-gas retrievals: a new approach based on geometry-dependent Lambertian equivalent reflectivity applied to OMI algorithms, Atmospheric Measurement Techniques, 10, 333–349, doi:10.5194/amt-10-333-2017, 2017.

Zhou, Y., Brunner, D., Boersma, K. F., Dirksen, R., and Wang, P.: An improved tropospheric $NO_2$ retrieval for OMI observations in the vicinity of mountainous terrain, Atmos. Meas. Tech., 2, 401–416, doi:10.5194/amt-2-401-2009, URL https://www.atmos-meas-tech.net/2/401/2009/, 2009.

---

## Author Comment (AC2) · 9 Nov 2018

**Evaluation of version 3.0B of the BEHR OMI NO$_2$ product**
**Response to Anonymous Referee #2**

Joshua L. Laughner, Qindan Zhu, and Ronald C. Cohen

November 8, 2018

We thank the reviewer for their positive comments. We acknowledge that there is a large volume of information presented here, and believe that the suggestions we have incorporated should help make the major points more apparent to the reader.

Responses to specific comments follow. The reviewer's comments will be shown in red, our response in blue, and changes made to the paper are shown in black block quotes. Unless otherwise indicated, page and line numbers correspond to the original paper. Figures, tables, or equations referenced as "R$n$" are numbered within this response; if these are used in the changes to the paper, they will be replaced with the proper number in the final paper. Figures, tables, and equations numbered normally refer to the numbers in the original discussion paper.

Section 4. The comparison of VCD should include some scatterplots. Presenting the comparison as a table (Table 2) only is a bit difficult to follow. Especially, the results with separated Pandora and aircraft VCDs should be shown. Some of these plots could go in the supplementary material.

Scatterplots with separate aircraft and Pandora data were already included in the supplement. Figures S1-S6 show separate aircraft and, where available, Pandora comparisons for each of the 6 campaigns used for validation. We have added a figure showing scatter plots for the combined aircraft and Pandora data to the main paper (Fig. R1).

You use a quite precise spatial collocation criteria (pandora site within the OMI pixel, so no spatial smoothing in practice) and then you time average Pandora observations ±1 h from the OMI observation, which is quite large time frame. Can you open a little bit about this choice? How does the results change with a shorter time interval?

The ±1 h averaging is the same as done in Goldberg et al. (2017). Part of our reason for using that averaging window is to make our results comparable with that study. Given the average wind for cities where the DISCOVER-AQ campaigns took place (4.5 m/s, from WRF simulations), a shorter averaging time window, e.g. ±0.5 h, might make more sense as this means an airmass would travel ∼16 km, which is similar to the length of an OMI pixel at nadir in either dimension (13 or 24 km). However, no slope changes by > 0.05 (max ∼ 8%) using the shorter window, so we prefer to be consistent with Goldberg et al. (2017) so that readers can compare our results. We have added a sentence to sect. 4 explaining this:

[Figure]

Figure R1: Scatter plots of the BEHR v2.1C, v3.0B (M), and (where available) v3.0B (D) VCDs against coincident aircraft and Pandora VCDs. All Pandora VCDs are used for these plots. Each panel is one campaign: the **(a)** Maryland, **(b)** California, **(c)**, Texas, and **(d)** Colorado DISCOVER-AQ campaigns. For slopes, see Table 3; for intercepts and $R^2$ values, see Table S3.

"As stated in sect. 2.3, we average all data within 1 hour of OMI overpass (i.e. 13:30 local time $\pm 1$ h) to be consistent with Goldberg et al. (2017). A shorter averaging window ($\pm 0.5$ h) was tested; the maximum effect on the slope was $\sim 8\%$ with most of the "matched" data showing differences of $\leq 5\%$ and the "all" data changing by $\leq 3.5\%$ in all but one case."

You mention several times in the text that changes in emission information as input have a role in the discrepancies you observe between different versions of the algorithm. Could you actually show them? For example, plotting or mentioning the quantitative the emission changes over the areas of study.

While the emissions files used in the original BEHR product are no longer available, we have reproduced their likely value based on the description in Russell et al. (2011) and Russell et al. (2012) and added a figure to the supplement (Fig R2).

Fig. 1: the plot is quite small, it could be a bit bigger so that the the different lines can be better separated? Also, the legend can be shown only once.

We have done our best to enlarge this figure within the space constraints of the AMT template.

P12 L11 only Pandora data that has a coincident aircraft profile is include -¿ only Pandora

[Figure]

Figure R2: Absolute **(a)** and percent **(b)** change in WRF-Chem NO emissions between the EPA NEI 2005 inventory, unscaled, and the EPA NEI 2011 inventory, scaled to 2012 levels.

data that have a coincident aircraft profile are included
   Corrected, thank you.

   Conclusions P17 L12 I think you refer to Sentinel 4, as Sentinel 5 is not planned on a geostationary orbit.
   Yes, thank you for the correction.

   Figure 1 and 3: the different panels would benefit from a title mentioning the different sites considered in the plot
   Added.

**References**

Goldberg, D. L., Lamsal, L. N., Loughner, C. P., Swartz, W. H., Lu, Z., and Streets, D. G.: A high-resolution and observationally constrained OMI NO$_2$ satellite retrieval, Atmos. Chem. Phys., 17, 11 403–11 421, doi:10.5194/acp-17-11403-2017, URL https://doi.org/10.5194/acp-17-11403-2017, 2017.

Russell, A. R., Perring, A. E., Valin, L. C., Bucsela, E. J., Browne, E. C., Wooldridge, P. J., and Cohen, R. C.: A high spatial resolution retrieval of NO$_2$ column densities from OMI: method and evaluation, Atmos. Chem. Phys., 11, 8543–8554, doi:10.5194/acp-11-8543-2011, URL https://doi.org/10.5194/acp-11-8543-2011, 2011.

Russell, A. R., Valin, L. C., and Cohen, R. C.: Trends in OMI NO$_2$ observations over the United States: effects of emission control technology and the economic recession, Atmos. Chem. Phys., 12, 12 197–12 209, doi:10.5194/acp-12-12197-2012, 2012.